# A Big World in Small Grain: A Review of Natural Milk Kefir Starters

**DOI:** 10.3390/microorganisms8020192

**Published:** 2020-01-30

**Authors:** Fatemeh Nejati, Stefan Junne, Peter Neubauer

**Affiliations:** Chair of Bioprocess Engineering, Institute of Biotechnology, Faculty III Process Sciences, Technische Universität Berlin, Straße des 17. Juni 135, 10623 Berlin, Germany

**Keywords:** lactic acid bacteria, milk kefir, consortia, identification, microbial interactions

## Abstract

Milk kefir is a traditional fermented milk product whose consumption is becoming increasingly popular. The natural starter for kefir production is kefir grain, which consists of various bacterial and yeast species. At the industrial scale, however, kefir grains are rarely used due to their slow growth, complex application, bad reproducibility and high costs. Instead, mixtures of defined lactic acid bacteria and sometimes yeasts are applied, which alter sensory and functional properties compared to natural grain-based milk kefir. In order to be able to mimic natural starter cultures for authentic kefir production, it is a prerequisite to gain deep knowledge about the nature of kefir grains, its microbial composition, morphologic structure, composition of strains on grains and the impact of environmental parameters on kefir grain characteristics. In addition, it is very important to deeply investigate the numerous multi-dimensional interactions among different species, which play important roles on the formation and the functionality of grains.

## 1. Introduction

Milk kefir is an ancient fermented milk drink that originates from the Caucasus. It is widely used in human nutrition due to its health-promoting properties. Traditionally, kefir is produced by fermenting milk with kefir grains, which consist of a mixture of microbial species. Most of the milk kefir grains’ habitants belong to the group of lactic acid bacteria (LAB), but kefir also contains yeasts and acetic acid bacteria (AAB). Depending on its age, the resulting kefir drink is typically acidic, of strong taste, partially viscous and fluffy [1].

For centuries, many health benefits were attributed to kefir, it was even consumed as a natural medicine [2]. Nutritional and medicinal properties of kefir have been in the focus of many scientific studies for decades. During milk fermentation by kefir grains, many functional compounds like bioactive peptides (e.g., with antihypertensive, antioxidative, antiallergenic, antitumor, antimicrobial, anti-inflammatory and cholesterol-lowering activities) [3,4], antimicrobial compounds (e.g., organic acids, alcohols, carbon dioxides and bacteriocins) and heteropolysaccharides (e.g., kefiran) with potential prebiotic activity are formed [5]. For a large number of kefir-isolated strains (e.g. *Lactobacillus (Lb.) kefiranofaciens*) and yeasts (e.g. *Kluyveromyces (Kl.) marxianus*), significant probiotic (probiotics are live microorganisms that, when administered in adequate amounts, confer a health benefit on the host) activities have been demonstrated in both in vitro and in vivo studies [6,7]. Based on these results, kefir is considered a “natural probiotic drink”, which underlines its uniqueness among other fermented dairy products [8]. However, kefir-based sensory and functional properties are prone to a drastic change in microbiota, the peptide/protein profile, metabolites, if defined mixed cultures are used as starter cultures instead of kefir grains [9,10]. This review aims to summarize recent studies on milk kefir with an emphasis on the microbial composition, fine structure analysis and multi-layer interactions and their roles in kefir grains. It further highlights research opportunities and important issues for future studies in connection to the functionality of natural microbial kefir consortia and their integrity. 

## 2. Microbial Structure of Kefir

The exact microbiological composition of kefir grains is still controversial. Up to 50 different bacterial and yeast species have been found in grain-based milk kefirs, which has been comprehensively reviewed elsewhere [11,12]. It seems that the geographical origin of the kefir samples (Table 1) and the cultivation conditions (e.g., different types of milk, temperatures, incubation times and ratios of grain and milk) may largely influence the microbial composition and dynamics of the kefir [13,14]. Nevertheless, the methods applied to identify this complex microbial community are not suitable to identify them correctly, as the results of microbial identification of the applied method influence by themselves (Table 1). For example, Kesmen and Kacmaz (2011) were able to identify *Lactococcus (Lc.) lactis, Leuconostoc (L.) mesenteroides* and *Lb. kefiri* as prevalent bacteria species with culture-dependent methods, while PCR denaturing gradient gel electrophoresis (DGGE) as a culture-independent method identified *Lb. kefiranofaciens* and *Lc. lactis* as prevalent [15].

Combinations of both, culture-dependent and rather traditional culture-independent methods, e.g., DGGE of PCR amplicons of rRNA-targeted gene regions, were the most common methods for the identification of kefir microbiota for many years [16]. Later, however, studies have shown that these combinations have limitations and drawbacks to accurately assess microbial communities [17,18]. For example, the V3 region of the 16S rRNA gene, which was widely used for the identification by rRNA-PCR-DGGE [15], cannot separate sufficiently closely related species like *Lb. kefiri*, *Lb. buchneri*, *Lb. sunkii* and *Lb. otakiensis* [18,19]. In contrast, newer identification techniques, like whole metagenome shotgun sequencing, provide more detailed information about the overall microbial structure, in particular for species of low abundance. These methods were able to provide a broader view on the microbial composition and population dynamics of kefir [20,21,22].

Although it seems that there is a big difference in the microbial composition among kefirs of different origins, the microbial composition, or at least dominant species, of kefir does not necessarily have to be complex. As an example, Wang et al. showed that *Lb. kefiranofaciens* is the only dominant bacterial species in Tibetan milk kefir grains independent of kefir production conditions by applying DGGE and metagenomic analysis [25]. According to several studies, just a few species, like *Lb. kefiranofaciens* (both subsp. *kefiranofaciens* and subsp. *kefirgranum*) and *Lb. kefiri* are ubiquitous, in which *Lb. kefiranifaciens* is homofermentative (lactic acid as the main end product), while *Lb. kefiri* is heterofermentative (producer of lactic acid, acetic acid, and carbon dioxide) [11,24]. The ratio of homofermentative to heterofermentative LAB has also been considered as a sensitive parameter in studies about microbial populations in kefir and its grain stability. Takizawa et al. observed that 90% of the total population of a kefir consisted of homofermentative bacteria (mainly of *Lb. kefiranofaciens*) [26], while Vardjan et al. found a ratio of homofermentative (*Lb. kefiranofaciens* subsp. *kefirgranum*) to heterofermentative (*Lb. kefiri* and *Lb. parakefiri*) species of about 1:1, which remained stable over four months [27]. The results of such culture-dependent studies, however, highly depend on the medium and conditions during species isolation. Additionally, plate counting for the quantification of viable cells is usually prone to a high variability.

## 3. Microbiota vs. Functional Properties

The single knowledge about the microbial composition of a community is not enough to understand the way each species or strain contributes to the formation of functional properties. Microbial communities are usually known as stable systems with mutual interactions between the different metabolic networks [8]. In order to achieve similar features in synthetic microbial consortia for industrial application, it is necessary to gain a deeper understanding of the allocated functions within the consortia and of the relation between microbiota and process performance.

‘Omics’ technologies (i.e., genomics, transcriptomics, proteomics and metabolomics) are powerful approaches to characterize the behavior of complex food consortia during the course of a fermentation [28,29]. Applying these comprehensive analyses allows the identification of the microorganisms(s) that play leading roles within the community [30,31]. For example, by the combination of metagenomics (to study the microbial dynamics) and metabolomics (to monitor the development of flavor compounds), it was observed that *Lb. kefiranofaciens* was the dominant microorganism in the early stages of milk kefir fermentation, while *L. mesenteroides* became more prevalent in later fermentation stages, which was correlated to concentration changes of volatile compounds [23]. 

Although there are still some limitations and challenges in design, application and data interpretation of omic-based analyses [32], it is obvious that they hold great potential to improve our understanding of microbial communities. The results of such studies are very helpful for the selection of most suitable strains to design artificial consortia or functional mixed starters [30].

## 4. Generation of Kefir Grains

The physical structure of kefir grains and the arrangement of microbiota within this structure is an important issue for the understanding of the microbiota’s functions and potential interactions within the consortia. Kefir grains exhibit an irregular cauliflower-like shape which consists of numerous hollow globular structures with a usual diameter of 2.0 to 9.0 mm, whereby the globules form a polyhedral network structure [33]. The matrix is composed of the exopolysaccharide kefiran, proteins, microbial cell debris and other materials, which were not specified so far [33,34]. The arrangement of microbiota on or in this structure is still a matter of research. Although some studies show that the microorganisms occupy all interior and exterior surfaces of grains [34,35], bacteria are hardly observed on the outer surfaces of grains, but only embedded in the fibrillar matrix near the surface [36]. It is very likely that the variations in grain cultivation conditions and environmental parameters caused these different observations about the arrangement of microbiota. Additionally, cell sizes and chain lengths can differ, in respect to physiological stages or external stresses (e.g., cultivation conditions and limitation of available nutrients), which may lead to false interpretations of microscopy data of microbial communities. For instance, *Lb. kefiranofaciens* was observed in two distinct morphotypes of short (3.0 µm in length) and long (10.0 µm in length) rods that have colonized either on the outer surfaces or inner surfaces of kefir grains [25].

It is unclear what causes the microbiota of kefir to form such a stable consortium that maintains its functionality for an infinite time. So far, all attempts to generate de novo kefir grain in any fermentation of mixtures of pure starter cultures have failed [37]. 

There are a few hypotheses, however, about the mechanisms involved in the formation of grains. It is assumed that the initial auto- and co-aggregation of lactobacilli and yeasts are the main initiating phenomena of the formation of small granules [11]. According to Wang et al., grain formation begins with self-aggregation of *Lb. kefiranofaciens* and *Kazachstania (Ka.) turicensis* [35]. Then, biofilm producing species like *Lb. kefiri* attach to the surface of the granules and co-aggregate with other microorganisms and milk components to form larger granules and probably kefir grains. Recently, the role of AAB like *Acetobacter (A.) orientalis* has been studied [34]. It seems that LAB and AAB are responsible for polysaccharide production and biofilm formation, while yeasts play a role in the evolvement of complex networks between the three microbes [34].

It is worth to mention that the occurrence of such inter-microbial interactions is highly strain specific. For example, it was observed that only six out of 20 *Lb. kefiri* strains were able to co-aggregate with *Saccharomyces (S.) lipolytica* strains, albeit all strains were isolated from kefir [38].

## 5. Inter-Microbial Interactions

Compared to a single taxon, microbial assemblages have proven to be highly resilient under adverse conditions and flexible in terms of substrate conversion [39]. These outstanding properties can be due to the participation of different microorganisms, wherein each of them carries its own specific genetic material and shares its metabolic features with others in the community. Although strong interaction is essential for achieving robustness in many ecological systems, the complexity increases with the number of microorganisms that are involved in a community [40]. Thus, studies become more difficult. In any kind of microbial community, microbial interactions can be studied from two aspects: (1) The nature of interactions and (2) the participating members.

### 5.1. Nature of Interactions

Species, as members of a community, typically influence each other’s growth and metabolism in several modes [41]. Interactions can be classified in two groups: (i) Direct, through physical contacts, and (ii) Indirect, either through the excretion of signal molecules (inhibitors/stimulators) or changing environmental growth factors (like pH and gas composition) [42].

Direct or physical interactions: In many cases, direct interactions and cell to cell attachments enable microorganisms to work cooperatively and form complex structures like a biofilm [43,44], which usually leads to an increased viability of the microbial community members and their resistance to stresses. For example, *S. cerevisiae* can significantly enhance the viability of the probiotic strain *Lb. rhamnosus* HN001 under acidic conditions [45]. Microscopic observations revealed that this effect is due to the direct cell to cell contact and subsequent co-aggregation that is mediated by yeast cell wall polysaccharides and bacterial cell surface proteins [38,46]. Mendes et al. reported the direct interaction between *S. cerevisiae* and *Lb. delbrueckii* as a strategy to cope with the adverse conditions at a low pH and in the presence of ethanol [47]. Cheirslip et al. showed the importance of the physical interactions between two kefir inhabitants, *S. cerevisiae* and *Lb. kefiranofaciens*, for enhanced production of exopolysaccharide kefiran by *Lb. kefiranofaciens* [48]. Exopolysaccharide kefiran is one of the main components of milk kefir grain. 

Indirect interactions: Primary and secondary extracellular metabolites have significant impacts on other partners of the community. Johansen et al. reviewed the functions of several quorum sensing (QS) (mediation of microbial cell–cell communication by secretion or recognition of small signal molecules) systems to regulate microbial traits in food-related communities and the potential effects on the quality of fermented products [49]. QS has been studied for several microorganisms involved in the production of fermented vegetables, sourdough, wine and some dairy products [50,51,52,53]. Although there are recent studies about the analysis of the metabolites profile of milk kefir [54,55,56], there is no thorough study to investigate the signaling functions of such metabolites on kefir-related species.

### 5.2. Participating Members

Relevant types of interactions within a kefir community can be categorized as follows:

• Yeast–Bacteria Interactions

The interaction between yeasts and LAB is central in a wide range of fermented foods, in particular in kefir [44]. Both groups of microorganisms naturally support each other in different ways:Assimilation of lactic acid: One interesting mechanism of interaction between yeasts and LAB is conducted in the presence of lactic acid assimilating-yeasts. Accumulation of lactic acid injures and kills LAB even when the pH of the culture is maintained by the addition of alkaline solutions [57]. However, lactic acid can be consumed as a carbon source by non-lactose-consuming yeasts, such as *S. cerevisiae*, which results in an increased pH and a prolonged growth of lactobacilli. This cooperation has been reported to strongly enhance production of the capsular kefiran by *Lb. kefiranofaciens* [48,58].Production of CO_2_/removal of O_2_: Carbon dioxide can provide a suitable atmosphere (reduced oxygen and elevated CO_2_) to favor *Lactobacillus* spp. growth. Even though no studies are available about kefir-isolated microorganisms, studies about other communities and food-isolated microorganisms verify this interaction. Suharja et al. linked the enhanced viability of probiotic *Lb. rhamnosus* to oxygen scavenging activity of *S. cerevisiae* [59]. A similar mechanism has been observed between *Lb. sanfranciscensis* and *S. cerevisiae*, two isolates of sourdough microbiota [60].Providing nutrients to bacteria: Trophic interactions and exchange of metabolites (cross-feeding) enable multiple groups of microorganisms to survive on limited resources. Yeast species have been shown to serve bacteria by providing vitamins, growth factors and essential amino acids [61]. Sadie et al. observed that *Zygotorulaspora florentina* excretes essential amino acids that support *Lb. nagelii* growth when they are co-cultivated, but not if they are cultivated as monoculture [62].

For exploring details of metabolite cross-feeding between *S. cerevisiae* and two species of LAB (*Lb. plantarum* or *Lc. Lactis*) in model systems, Ponomarova et al. used combined metabolomics and genetic tools [61]. They demonstrated how nitrogen overflow by yeast contributes to the emergence of mutualism with *Lc. Lactis*. Mutualism between *Lc. lactis* and *S. cerevisiae* easily emerges when lactose is the main carbon source. This finding highlights again the fact that the composition of the growth medium has an important role on the formation of inter-species interactions.

• Bacteria–Bacteria Interactions

Food-related bacteria–bacteria interactions have not been studied to the same extent as yeast–bacteria interactions. There are informative studies on bacteria–bacteria interactions between bacterial species of yogurt, *Lb. delbrueckii* subsp. *bulgaricus* and *Streptococcus (S.) thermophilus*, which are known for their protocooperative and symbiotic interactions [41]. When grown together, *Lb. delbrueckii* subsp. *bulgaricus* encoding the hydrophobic di/tripeptides Dpp transport system, which is complemented by the general di/tripeptide DtpT transporter system in *S. thermophilus*. This interaction results in the utilization of more peptides by both bacteria [63]. In a recent study, interactions between several kefir bacterial species (e.g., *Lb. kefiranofaciens*, *Lb. kefir*, *Lc. lactis*. *A. fabarum* and *L. mesenteroides*) in pairwise combinations have been the topic of investigation and a more detailed layer has been added to the knowledge of kefir-related microbiota interactions [54]. According to this study, *Lb. Kefiranofaciens*, a ubiquitous strain of kefir microbiota, suppressed growth of its direct competitor *Lb. kefiri*, while promoting growth of *L. mesenteroides* and having no effect on *Lc. lactis* and *A. fabarum*.

• Yeast–Yeast Interactions

Communication among yeasts through QS is not as well documented. In some food ecosystems like wine and sourdough, some studies have been carried out to identify metabolites and conditions involved in QS communication in the model yeast *S. cerevisiae* [53]. These show environmental factors, like the nitrogen content in the medium, cell density, aerobic/anaerobic conditions and ethanol content affect profoundly the production of QS-related molecules by *S. cerevisiae* [64]. For example, yeasts QS-related molecules like aromatic alcohols (e.g., tryptophol, tyrosol and 2-phenylethanol) are secreted at highest rates when the ammonium sulfate concentration is below 50 μM. These rates are reduced when the ammonium sulfate concentration raises above 500 μM [49]. Interestingly, such QS-related aromatic molecules are applied also as antioxidant or antimicrobial agents and are used in quality control assessments [64].

Some strains of *S. cerevisiae* secrete peptides that inhibit the growth of some non-*Saccharomyces* strains, such as *Kl. marxianus* [65]. This characteristic is reported to be highly strain-dependent [66]. As *S. cerevisae*, *Kl. marxianus* and *Kazachstania* spp. (*Kz. turicensis*, *Kz. unispora* and *Kz. exigua*) are important yeast species of kefir microbial community [18,44,54], unravelling of the interactions among these species and the relation to kefir quality and grains functionalities are very important.

## 6. Effects of External Parameters on Kefir Robustness and Integrity

Environmental changes during any fermentation lead to harsh stress on microorganisms, so that survival under the newly developed conditions depends highly on stress response mechanisms. The inhabitants of kefir grains are also confronted with several stress factors during milk fermentation (such as high acidity, rapid temperature fluctuations, limitation of nutrients, presence of antimicrobial compounds, etc.) and not only during a single fermentation, but also during successive fermentations. Each can affect the population dynamics drastically [67]. Vardjan et al. [68] however, demonstrated that the prevailing lactobacilli and yeasts in kefir grains and kefir beverages are stable during ten weeks of propagation. It has not been investigated decently which parameters are responsible for the protection of the kefir microbiota against successive environmental stresses. Investigation of *Lb. kefiranofaciens* M1 stress adaptation under lethal and sub-lethal levels of heat, cold, acid and bile salt stresses shows up-regulating of several classes of proteins belong to carbohydrate metabolism (TpiA, ENO and GPDH), pH homeostasis (AtpA and AtpB), stress response proteins (DnaK and GroEL) and translation-related protein (Rps2) [69]. Despite the importance of this subject, the knowledge how about kefir grain microbiota cope with stresses is very limited. What are the roles of inter-species interactions in stress adaptation, is something that is worth being studied. 

## 7. Conclusions

The purpose of this review was to summarize the latest findings in grain-based milk kefir studies and lighten up possible fundamental research that is required to define parameters of robustness and integrity of kefir grains. This knowledge can be helpful for developing fully functional artificial starter cultures for increased quality attributes and consumer acceptance for industrial application. 

## Figures and Tables

**Table 1 microorganisms-08-00192-t001:** Microbial composition of kefir samples of different geographical locations.

Geographic Location of Studied Sample	Method of Analysis	Identified Bacteria or Yeasts	Reference
France, Ireland and the United Kingdom	Metagenomics (16S rRNA and ITS sequencing)	*Lb. kefiranofaciens*, *Leuconostoc* spp., *Lb. helveticus*, *A. pasteurianus, Saccharomyces* spp. and *Kazachstania* spp.	[23]
Belgium	Metagenomics (16S and 26S rRNA sequencing)	*Lb. kefiranofaciens* or *Lc. lactis*, *Lb. kefiri, Acetobacter* spp. and *Enterobacter* spp., *Kl. marxianus*, *Kz. exigua* and *Nauvomozyma* spp.	[22]
Malaysia	Metagenomics (16S rRNA sequencing)	*Lb. kefiranofaciens* and *Lb. kefiri*	[20]
Italy	Metagenomics (16S rRNA and 26S rRNA sequencing)	*Lb. kefiranofaciens* as dominant and *Lb. kefiri*, *Enterococcus* spp., *Lc. lactis and Acetobacter* spp. as subdominant bacteria, *Dekkera anomalus*, *Kz. exigua*, *S. cerevisiae*	[18]
Brazil	PCR-DGGE	*Lb. kefiranofaciens* and *L. kefiri*	[17]
	Metagenomics (16S rRNA sequencing)	*Lactobacillus* spp., such as *Lb. kefiranofaciens* subsp. *kefirgranum*, and subsp. *kefiranofaciens*, *Lb. kefiri*, *Lb. parakefiri*, *Lb. parabuchneri*, *Lb. amilovorus*, *Lb. crispatus* and *Lb. buchneri*	
Turkey	Metagenomics (16S rRNA sequencing)	*Lb. kefiranofaciens*, *Lb. buchneri*, *Lb. helveticus*	[19]
Turkey	PCR-DGGE	*Lb. kefiranofaciens*, *Lb. kefiri*, *Lb. buchneri*, *Lb. sunkii*, *Lb. otakiensis*	[15]
	Culture dependent	*Lc. lactis*, *L. mesenteroides*, *Lb. kefiri*	
Russia	Culture dependent	*Lb. kefiranofaciens*, *Lb. kefiri*, *Lb. parakefiri*, *Lc. lactis* and *Leuconostoc* spp.	[24]

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
