# Peer review of "A Big World in Small Grain: A Review of Natural Milk Kefir Starters"

_microorganisms, 2020, doi:10.3390/microorganisms8020192_

Round 1
Reviewer 1 Report
This article summarizes current understanding of the kefir microbiome and relevant ecology pertaining to structure and form of the kefir grain, and identifies clear areas of research that need to be undertaken, provide good direction for research needed in this field. However, the numerous errors in style and English (over 50 small mistakes) preclude any serious consideration for this manuscript in its current state, and the corrections are too extensive to list here. It appears that this article lacked proof-reading, as many of these errors are easily caught, in some case by spellcheck functions.
Author Response
|
Comment |
Response |
|
This article summarizes current understanding of the kefir microbiome and relevant ecology pertaining to structure and form of the kefir grain, and identifies clear areas of research that need to be undertaken, provide good direction for research needed in this field. However, the numerous errors in style and English (over 50 small mistakes) preclude any serious consideration for this manuscript in its current state, and the corrections are too extensive to list here. It appears that this article lacked proof-reading, as many of these errors are easily caught, in some case by spellcheck functions.
|
All Language mistakes and errors were corrected. The manuscript was corrected by several people, partly native speakers. |
Reviewer 2 Report
Comments on the work:
A big world in small grain: A review on natural kefir starter
I recommend the manuscript to be published in Microorganisms after minor revision.
In the continuation you can find some corrections and comments.
P – page
L – line
Keywords
P2 L21: instead of: “microbial interactions”, it should be: “Microbial interactions”.
Microbial structure of kefir
P3 L10: instead of: “For example, Kesmen and Kacmaz (2011) by using culture-dependent method could identify L. lactis, Leuconostoc (leuc.) mesenteroides, and Lb. Kefiri as prevalent bacteria species, while by applying culture-independent method (PCR- denaturing gradient gel electrophoresis (DGGE)) Lb. kefiranofaciens and L. lactis, respectively in kefir grain and beverage, were identified as prevalent species [15].”, shouldn’t it be?: “For example, Kesmen and Kacmaz (2011) could identify by using culture-dependent method L. lactis, Leuconostoc (leuc.) mesenteroides, and Lb. Kefiri as prevalent bacteria species, while by applying culture-independent method (PCR- denaturing gradient gel electrophoresis (DGGE)) Lb. kefiranofaciens and L. lactis were identified as prevalent species in kefir grain and beverage [15].”.
Table 1: Review the text in the column of identified bacteria or yeast (italic or not, especially sp.).
Microbiota vs functional properties
P7 L101-104: Why do you mention the study of Laureys and De Vuyst [32]? they were performing fermentation using water kefir grains, which can certainly not be compared to milk kefir grains, because of different microbial composition. Remove this sentence because it can be misleading for the reader.
Generation of kefir grains
P7 L113: instead of: “…consists of numerous hollow globular structures with a diameter of usually 2.0 to 9.0 mm,…”, it should be: “…consists of numerous hollow globular structures usually with a diameter of 2.0 to 9.0 mm,…”.
P8 L131: instead of: “According to Wang et al. that grain formation begins with…”, it should be: “According to Wang et al. grain formation begins with…”.
P8 L134: instead of: “Recently, also the role of acetic acid bacteria like Acetobacter (A.) orientalis has been studied [34].”, it should be: “Recently, the role of acetic acid bacteria like Acetobacter (A.) orientalis has been studied [34].”.
Inter-microbial interactions
P9 L154: instead of: “…and ii. Indirect, through producing…”, it should be: “…and ii. indirect, through producing…”.
P10 L178: instead of: “…wide range of fermented foods and especially kefir…”, it should be: “…wide range of fermented foods especially kefir…”.
Conclusion
P13 L254: instead of: “This knowledge is can be helpful for developing fully…”, it should be: “This knowledge can be helpful for developing fully…”.
Author Response
|
Comments |
Response |
|
Keywords P2 L21: instead of: “microbial interactions”, it should be: “Microbial interactions”. Microbial structure of kefir P3 L10: instead of: “For example, Kesmen and Kacmaz (2011) by using culture-dependent method could identify L. lactis, Leuconostoc (leuc.) mesenteroides, and Lb. Kefiri as prevalent bacteria species, while by applying culture-independent method (PCR- denaturing gradient gel electrophoresis (DGGE)) Lb. kefiranofaciens and L. lactis, respectively in kefir grain and beverage, were identified as prevalent species [15].”, shouldn’t it be?: “For example, Kesmen and Kacmaz (2011) could identify by using culture-dependent method L. lactis, Leuconostoc (leuc.) mesenteroides, and Lb. Kefiri as prevalent bacteria species, while by applying culture-independent method (PCR- denaturing gradient gel electrophoresis (DGGE)) Lb. kefiranofaciens and L. lactis were identified as prevalent species in kefir grain and beverage [15].”. Table 1: Review the text in the column of identified bacteria or yeast (italic or not, especially sp.). Microbiota vs functional properties P7 L101-104: Why do you mention the study of Laureys and De Vuyst [32]? they were performing fermentation using water kefir grains, which can certainly not be compared to milk kefir grains, because of different microbial composition. Remove this sentence because it can be misleading for the reader. Generation of kefir grains P7 L113: instead of: “…consists of numerous hollow globular structures with a diameter of usually 2.0 to 9.0 mm,…”, it should be: “…consists of numerous hollow globular structures usually with a diameter of 2.0 to 9.0 mm,…”. P8 L131: instead of: “According to Wang et al. that grain formation begins with…”, it should be: “According to Wang et al. grain formation begins with…”. P8 L134: instead of: “Recently, also the role of acetic acid bacteria like Acetobacter (A.) orientalis has been studied [34].”, it should be: “Recently, the role of acetic acid bacteria like Acetobacter (A.) orientalis has been studied [34].”. Inter-microbial interactions P9 L154: instead of: “…and ii. Indirect, through producing…”, it should be: “…and ii. indirect, through producing…”. P10 L178: instead of: “…wide range of fermented foods and especially kefir…”, it should be: “…wide range of fermented foods especially kefir…”. Conclusion P13 L254: instead of: “This knowledge is can be helpful for developing fully…”, it should be: “This knowledge can be helpful for developing fully…”.
|
All corrections have been performed and highlighted in yellow. |
Reviewer 3 Report
See attachment

Author Response
|
Comment |
Response |
|
This paper is a review on natural kefir starter. Relevance of the paper is linked to the increasing attention on kefir as a “natural” functional food and its health benefits. Most aspects on research about kefir have been included in the paper highlighting the need to further investigate in order to gain knowledge on them. The review compiles information from different studies but results are not discussed/digested in depth for a significant contribution to improve the current knowledge on kefir microbiology.
|
So far, the main focus on kefir studies was limited to two topics: 1. investigation of the microbial composition and 2. health properties of kefir. The industry has failed, however, to design authentic kefir starters so far. This is due to a lack of detailed studies addressing the formation and structure of specific microbial communities, e.g. the aggregation process, the evaluation of the robustness of kefir grains, and interactions between species. In this review, we aim to clarify the importance of these fields and to reorient future kefir studies and the aim was not merely contributing to the improvement of kefir microbiology knowledge. The aims of this review were highlighted in lines 47 to 51 (Introduction). |
|
As a review it is not exhaustive enough and some important data on microbial composition of kefir are missing i.e. Bifidobacteria. Please see the papers below as reference: - Laureys D, De Vuyst L. Microbial species diversity, community dynamics, and metabolite kinetics of water kefir fermentation. Appl Environ Microbiol. 2014;80(8):2564–2572. doi:10.1128/AEM.03978-13
- A. Gulitz, J. Stadie, M.A. Ehrmann, W. Ludwig and R.F. Vogel. Comparative phylobiomic analysis of the bacterial community of water kefir by 16S rRNA gene amplicon sequencing and ARDRA analysis. Journal of Applied Microbiology 2013; 114, 1082—1091. doi:10.1111/jam.12124 |
The aim of this review is not to extensively discuss the microbial composition of kefir (and not at all about the species that are in minority). As it was mentioned in the review, there are several other reviews that comprehensively presented that microbial profile, they have been referenced (Lee et al., 2018 and Prado et al. 2015). In addition, the study of Laureys and De Vuyst (2014) included in the first version of the manuscript, was removed as it was about water kefir and not milk kefir, according to the suggestion of another reviewer. In the current version of the paper, we have made it clearer that we are focusing solely on milk kefir and excluding water kefir.
|
|
Page 3. Footnote. Definition of Probiotics is not correct as is “Probiotics are live microorganisms which, when administered in adequate amounts, confer a health benefit on the host by settling in the gastro-intestinal tract”.
|
This was corrected.
|
|
Table 1. Please revise. It has many mistakes i.e. 26s, And, …. missing cursive…. Use S. cerevisiae and not Sac. cerevisiae; K. exigua and not Kaz. exigua |
The corrections were performed in Table1. We used Kz. for Kazachstania and Kl. for kluyveromyces in this review. |
|
Page 6. Lines 77-78. The part of the sentence “independent of culture conditions” probably refers to the “kefir production conditions”. It should be clarified. Line 78. According to the results of several studies….. Line 79. ….. subsp. kefirgranum Line 82. …… LAB has been also…… Line 84. Please provide some more information about the study to understand the different proportions Line 94. Omics technologies “are” powerful….. Line 96. Applying of……delete “of” Line 98. ….study of ….. delete “of” Line 100. leuc. mesenteroides change by “L. mesenteroides” Line 103. …. in compare to yeasts … change by compared to yeasts or in comparison to yeasts Lines 116, 120, 193. Change “microflora” by “microbiota” Line 121. …. experiencing of external stresses … Revise the sentence. Line 129. However (.) Line 145. …shares its metabolic….. Line 152. ….. , as members of a community, … Line 174. “descent” Please clarify what is meant with this word. Line 187….. to enchance… Line 189. “Lactobacillus sp.” the author probably means different Lactobacillus species. In this case it should be “Lactobacillus spp.” Line 192. S. cervisiae Line 194. Change to “Providing nutrients to bacteria” Lines 204-205. Please rewrite to clarify what is meant in these sentences Line 211. Change to Lb. delbrueckii subsp. bulgaricus Line 215. lactis Line 219. Delete “it” Line 220. …. have been carried “out” Line 242. “It has not studied decently”…. Please rewrite this sentence Line 249. “adaptation” Line 254. This knowledge (is) can be helpful …. delete “is”
|
All corrected and highlighted. |
Round 2
Reviewer 3 Report
Just some minor corrections:
Line 84, "conditions"
Line 191, I would suggest to use any of these words "thorough, rigurous, in-depth" instead of "decent"
Line 208, Lactobacillus "spp." and not "ssp."